# Polyubiquitination and SUMOylation Sites Regulate the Stability of ZO-2 Protein and the Sealing of Tight Junctions

**DOI:** 10.3390/cells11203296

**Published:** 2022-10-19

**Authors:** Misael Cano-Cortina, Lourdes Alarcón, Jael Miranda, Otmar Huber, Lorenza González-Mariscal

**Affiliations:** 1Department of Physiology, Biophysics and Neuroscience, Center for Research and Advanced Studies (Cinvestav), Ave IPN 2508, Mexico City 07360, Mexico; 2Institute of Biochemistry II, Jena University Hospital, Friedrich-Schiller-University Jena, Nonnenplan 2-4, 07743 Jena, Germany

**Keywords:** ZO-2, tight junction, ubiquitination, SUMOylation, epithelia

## Abstract

Tight junctions (TJs) regulate the transit of ions and molecules through the paracellular pathway in epithelial cells. *Zonula occludens* 2 (ZO-2) is a cytoplasmic TJ protein. Here, we studied the ubiquitination of hZO-2 employing mutants of SUMOylation site K730 present in the GuK domain and the putative ubiquitination residues K759 and K992 located at the GuK domain and proline-rich region, respectively. In immunoprecipitation experiments done with MDCK cells transfected with wild-type (WT) hZO-2 or the ubiquitination-site mutants hZO-2-K759R or -K992R, we observed diminished ubiquitination of the mutants, indicating that residues K759 and K992 in hZO-2 are acceptors for ubiquitination. Moreover, using TUBES, we found that residues K759 and K992 of hZO-2 are targets of K48 polyubiquitination, a signal for proteasomal degradation. Accordingly, compared to WT hZO-2, the half-life of hZO-2 mutants K759R and K992R augmented from 19.9 to 37.3 and 23.3 h, respectively. Instead, the ubiquitination of hZO-2 mutant K730R increased, and its half-life diminished to 6.7 h. The lack of these lysine residues in hZO-2 affects TJ sealing as the peak of TER decreased in monolayers of MDCK cells transfected with any of these mutants. These results highlight the importance of ZO-2 ubiquitination and SUMOylation to maintain a healthy and stable pool of ZO-2 molecules at the TJ.

## 1. Introduction

Epithelial cells have at the uppermost portion of their lateral membrane a cell–cell adhesion structure known as tight junction (TJ) [1] that regulates the transit of molecules through the paracellular pathway [2] and blocks the free diffusion of lipids and proteins within the membrane from the apical to the basolateral surface [3]. *Zonula occludens 2* (ZO-2) is a peripheral protein, a member of the MAGUK (membrane-associated guanylate kinase homolog) protein family that functions as a platform for the polymerization of the integral TJ proteins claudins into strands [4]. 

ZO-2 displays multiple protein–protein domains and motifs that allow this protein to function as a platform involved in diverse processes and signaling pathways. In addition, ZO-2 moves between the cytoplasm, the nucleus, and the cell borders due to various functional nuclear localization and exportation signals (for review, see [5]). These characteristics allow ZO-2 to perform multiple functions beyond those expected for a TJ protein. For example, at the nucleus, ZO-2 associates with transcription factors [6] and acts as a transcriptional repressor of genes involved in cell proliferation like cyclin D1 [7] or that are a target of oncogenic proteins like Yap [8,9] or the Wnt pathway [10,11]. ZO-2 also maintains the nuclear shape due to its interaction with proteins like lamin B and SUN-1 that link the nucleoskeleton to the cytoskeleton [12]. ZO-2 also regulates cell size [8] by acting as a platform of the Hippo pathway [8,13]. ZO-2 is a target of viral oncoproteins, including v-Src [14,15], the E4 region-encoded (E4-ORF1) of adenovirus type 9 that induces the aberrant sequestration of ZO-2 in the cytoplasm [16], and the E6 protein from high-risk human papillomavirus that delocalizes ZO-2 from the membrane to the cytoplasm and nucleus and stabilizes ZO-2 expression [17,18]. ZO-2 behaves as a tumor suppressor protein since its expression diminishes in a wide variety of carcinomas [19,20,21,22], and its absence in infants triggers progressive intrahepatic cholestasis that leads to hepatocellular carcinoma [23,24,25]. All these observations have prompted us to explore the mechanisms that regulate ZO-2 protein stability in epithelia.

The ubiquitin–proteasome system (UPS) is responsible for the non-lysosomal degradation of cytosolic, nuclear, and endoplasmic reticulum (ER)-residing proteins. In this process, proteins are tagged by the small protein ubiquitin which earmarks them for destruction in the 26S proteasome, an ATP-dependent protease complex (for review, see [26]). Conjugation of ubiquitin to a substrate requires the action of three enzymes: E1, a ubiquitin-activating enzyme; E2, a ubiquitin-conjugating enzyme; and E3, a ubiquitin-ligase enzyme that catalyzes the transfer of ubiquitin to the target protein destined for degradation. Ubiquitin is covalently bound to the target protein by its carboxy-terminal glycine, forming a link with the ε-amino group of lysine in the target protein. A similar linkage is also formed between the carboxy terminus of ubiquitin with the ε-amino group of lysine of another ubiquitin to form polyubiquitin chains. Since seven lysine residues are present in ubiquitin at positions 6, 11, 27, 29, 33, 48, and 63, each can function as an acceptor of another ubiquitin molecule. K48 and K11 are the most abundant ubiquitin chain types, and the former is the canonical signal that targets proteins to the proteasome for degradation while K11 is a potent degradation signal for particular proteins like those of the cell cycle and endoplasmic reticulum. The other types of homogeneous polyubiquitination serve a diverse role in the cell. Thus, K6 is involved in DNA repair, K27 in mitophagy, K29 in the degradation of amino-terminal ubiquitinated substrates and autophagy, K33 in kinase modification, and K63 in endocytosis, DNA repair, and macroautophagy. Likewise, monoubiquitination and multi-monoubiquitination serve other purposes, including endocytosis, aggregation, and macroautophagy (for review, see [27]).

In a previous study, we demonstrated that SUMOylation regulates the intracellular fate of ZO-2 [11]. Through an in silico search (SUMOsp 2.0 program; SUMOsp.biocuckoo.org/online.php), we identified six putative SUMOylation sites in hZO-2: K117, K730, K759, K889, K992 and K1003, and with a Ubc9 fusion directed SUMOylation assay found that of these residues only K730 is a target of SUMOylation and association to SUMO conjugating and deconjugating enzymes [11]. K759 and K992 residues of hZO-2 that did not behave as SUMOylation sites are now the subjects of the present study, as we observed that they also constitute putative ubiquitination sites.

Here, we have studied how the ubiquitination of hZO-2 is affected by the mutation of residues K730, K759, and K992 and the impact exerted on hZO-2 stability and TJ sealing in epithelial MDCK cells. First, we observed that K759 and K992 are K48 polyubiquitination targets, the canonical signal for proteasomal degradation. Accordingly, we found that in comparison to wild-type (WT) hZO-2, mutants K759R and K992R display an increase in half-life from 19.9 h to 37.3 h and 23.3 h, respectively. Instead, the hZO2 mutant in the K730 SUMOylation site displays increased ubiquitination and a decrease in half-life to 6.7 h. Furthermore, changing the ubiquitination/SUMOylation capacity of ZO-2 altered its interaction with other TJ proteins, as ubiquitin-deficient mutants displayed an increased interaction with occludin (K759R and K992R) but a diminished association to ZO-1, whereas the SUMOylation mutant K730R increased its interaction with ZO-1 and diminished its binding to occludin. These changes appear to affect TJ sealing since monolayers of MDCK cells transfected with any of these hZO-2 mutant constructs develop a lower peak of TER upon de novo TJ formation, thus suggesting that the proper ubiquitination/SUMOylation of ZO-2 is essential in order to maintain a stable pool of the protein and avoid the presence of defective molecules in the junction. 

## 2. Materials and Methods

### 2.1. Cell Culture 

Epithelial MDCK cells from the American Type Culture Collection (Manassas, VA, USA; CRL-2936) between 60th and 100th passages were grown as previously described [28]. 

### 2.2. cDNA Constructs

Generation of cDNA constructs of FLAG_3_-hZO-2 and the mutants FLAG_3_-hZO-2-K759R, -K992R, and -K730R was previously described by us [11]. The pcDNA3-HA-ubiquitin (HA-Ubi) construct was provided by Dr. Ivan Dikic (Goethe University Frankfurt/Main, Germany). 

### 2.3. Immunoprecipitations

Immunoprecipitations were done following a standard procedure previously described [10]. Transfected FLAG_3_-hZO-2 construct and the mutants FLAG_3_-hZO-2-K759R, -K992R, and -K730R were immunoprecipitated with 1 μg/μL of a mouse anti-FLAG M2 antibody (Cat. F1804, Sigma Aldrich, Saint Louis, MO, USA). For the experiments done to test the ubiquitination of ZO-2 mutants, 24 h after transfection, the cells were incubated for 2 h with 50 μM of PR616 (Cat. SI9619, Life Sensors, Malvern, PA, USA), a permeable inhibitor of ubiquitin/ubiquitin-like proteases that protects polyubiquitylated proteins from degradation [29].

### 2.4. Cellular Lysates, SDS-PAGE, and Western Blot

The cellular lysates, SDS-PAGE, and Western blot were done as previously reported [30]. For the blots, we employed mouse monoclonal antibodies against HA (Cat. H3663, Sigma Aldrich, St. Louis, MO, USA), FLAG (Cat. F1804; dilution 1:5000, Sigma-Aldrich, St Louis, MO, USA), occludin (Cat. 33-1500; dilution 1:1000, Invitrogen, Camarillo, CA, USA) and actin (generated and generously provided by Dr. Manuel Hernández, Cinvestav, Department of Cell Biology), and a rat monoclonal anti-ZO-1 (Cat. R26.4C, dilution 1:500; Developmental Studies Hybridoma Bank, University of Iowa, Iowa, IA, USA). As secondary antibodies, we employed peroxidase-conjugated goat anti-mouse IgG (Cat. 626520, dilution 1:10,000 Invitrogen, Camarillo, CA, USA) and anti-rat IgG (Cat. 81-9520, dilution 1:10,000, Invitrogen, Camarillo, CA, USA), followed by Immobilon chemiluminescence detection (Cat. WBKLS 0500, Darmstadt, Germany).

### 2.5. Isolation of K48 Polyubiquitinated hZO-2 with His-tagged K48-TUBES

MDCK cells were co-transfected with pcDNA3-HA-ubiquitin (HA-Ubi) plus Flag_3_-hZO-2-WT, or the ubiquitin mutants FLAG_3_-hZO-2-K759R and -K992R, or the negative control mutant FLAG_3_-hZO-2-K730R. After 24 h, the cells were incubated for 2 h with 50 μM of PR616. Then, the monolayers were washed twice with cold PBS and treated with ice-cold lysis buffer (100 mM Tris/HCl, pH 8.0, 0.15 M NaCl, 5 mM EDTA, 1% (*v*/*v*) NP40, and 0.5% (*v*/*v*) Triton X-100) with 50 μM of PR616 and the protease inhibitor cocktail Complete (Cat. 04906837001; Roche, Indianapolis, IN, USA). Next, cells were removed from the substrate with a rubber policeman and resuspended by pipetting. The lysate was clarified by centrifugation at 14,000× *g* for 20 min at 4 °C. Lysates supernatants were incubated overnight with His-tagged anti-K48 TUBES (Cat. UM405, Life Sensors, Malvern, PA, USA). Then, a sepharose-based Ni^2+^ chelate resin was employed to purify His-tagged proteins (Complete His-Tag Purification Column, Cat. 06781543001, Sigma Aldrich, St. Louis, MO, USA). The column was first equilibrated according to the manufacturer’s instructions with buffer A (50 mM NaH_2_PO_4_, pH 8.0, 300 mM NaCl). Then, the cell lysate was added to the His-tag column and incubated for 4 h at 4 °C with gentle rotation. His-tagged proteins were then eluted with buffer A with 250 mM imidazole pH 6.5. Eluted proteins were then subjected to an SDS-PAGE followed by a Western blot with mouse anti-Flag M2 (Cat. F1804; dilution 1:5000, Sigma-Aldrich, St Louis, MO, USA).

### 2.6. Analysis of hZO-2 Stability by the Cycloheximide Chase Assay

The analysis of hZO-2 stability was done as previously described [17]. Briefly, sparse cultures of MDCK cells were transfected with Flag3-hZO-2 WT, the ubiquitin mutants FLAG_3_-hZO-2-K759R and -K992R, or the SUMOylation mutant FLAG_3_-hZO-2-K730R. After the 6 h needed for the optimal expression of the transfected protein, the cells were incubated for different chase periods (0, 5, 10, 20, and 30 h) at 37 °C with CDMEM containing 30 μM cycloheximide (Cat. 1810, Sigma-Aldrich, St. Louis, MO, USA), an inhibitor of protein synthesis. Monolayers were then lysed with ice-cold lysis buffer (100 mM Tris/HCl, pH 8.0, 0.15 M NaCl, 5 mM EDTA, 1% (*v*/*v*) NP40, and 0.5% (*v*/*v*) Triton X-100) containing the protease inhibitor cocktail Complete (Cat. 04906837001; Roche, Indianapolis, IN, USA). Cells were detached from the substrate with a rubber policeman and resuspended by pipetting. The lysate was clarified by centrifugation at 14,000× *g* for 20 min at 4 °C. The supernatant was run in an SDS-PAGE followed by a Western blot with anti-Flag M2 (Cat. F1804; dilution 1:5000, Sigma-Aldrich, St Louis, MO, USA) and anti-actin antibodies. 

### 2.7. Measurement of Transepithelial Electrical Resistance (TER)

MDCK cells were plated on Transwell clear inserts (pore size 0.4 μm; Corning Inc., Cat. 3460, Corning, NY, USA) and one day later were transfected in DMEM without calcium and serum with Lipofectamine™ 2000 (Life Technologies, Cat.11668-019, Carlsbad, CA, USA) with the following constructs: Flag_3_-hZO-2 WT, the ubiquitination mutants FLAG_3_-hZO-2-K759R and -K992R, or the SUMOylation mutant FLAG_3_-hZO-2-K730R. After 6 h of incubation needed for the cells to start producing hZO-2 protein from the transfected constructs, the monolayers were switched to DMEM with 1.8 mM CaCl_2_ and bovine serum. After that, the value of TER was continuously measured from each insert in the automated cell monitoring system, CellZscope (nanoAnalytics GmbH, Münster, Germany), using the CellZscope software, version 1.5.0.

### 2.8. Paracellular Flux Assay

The paracellular flux assay was done as previously described [31]. For this assay, we added 200 μL of the tracer solution of 10 μg/mL FITC-Dextran of 10 kDa (Invitrogen, Cat. D1821, Eugene, OR, USA) or 70 kDa (Invitrogen, Cat. D1823, Eugene, OR, USA) to the apical side of confluent monolayers plated on Transwell inserts (Costar, Cat. 3470, Tewksbury, MA, USA). After incubation for 1 h at 37 °C, media from the upper and lower chambers were collected and the amount of FITC-Dextran was measured in a fluorometer (excitation 492 nm; emission 520 nm). This assay was done 18 h after MDCK monolayers were transfected with the following constructs: Flag_3_-hZO-2 WT, the ubiquitination mutants FLAG_3_-hZO-2-K759R and -K992R, or the SUMOylation mutant FLAG_3_-hZO-2-K730R. Time 0 corresponds to the initial 6 h recommended by the Lipofectamine™ 2000 manufacturer for the expression of the transfected construct. 

## 3. Results 

### 3.1. K759 and K992 Are Lysine Acceptors for Ubiquitination of ZO-2

An in silico analysis of the hZO-2 sequence done with Ubpred (www.ubpred.org: accessed on 15 March 2017) revealed that residues K759 and K992 are putative ubiquitination sites conserved in several species (Figure 1A,B). 

To examine if K759 and K992 are targets for ubiquitination, amino-terminal FLAG_3_-tagged hZO-2 (Flag_3_-hZO-2 WT) or the mutants FLAG_3_-hZO-2-K759R or -K992R were co-transfected with HA-ubiquitin (HA-Ubi) into MDCK cells. In addition, the mutant FLAG_3_-hZO-2K730R, which lacks a SUMOylation target residue, was also included as a negative control. Cell lysates were immunoprecipitated with anti-FLAG antibody or pre-immune serum, and the precipitates were analyzed on Western blots with anti-HA antibody. 

Figure 2 shows that compared to hZO-2-WT, the degree of ubiquitination is higher in the SUMOylation mutant FLAG_3_-hZO-2K730R and diminishes in hZO-2 ubiquitination mutants K759R and K992R. These observations suggest that K759 and K992 residues of hZO-2 are targets of ubiquitination and that the lack of the functional SUMOylation site K730 promotes the ubiquitination of hZO-2.

### 3.2. K759 and K992 Residues of hZO-2 Are Targets of K48 Polyubiquitination

K48-linked ubiquitin chains are the most abundant type of ubiquitin chains in mammalian cells, and in purified proteasomes, K48-ubiquitinated proteins are the most rapidly degraded (for review, see [27]). The proteasome recognizes K48-linked tetraubiquitin as the minimal targeting signal, and the binding strength increases as chain length augments up to octaubiquitin [32]. Therefore, we next employed Tandem Ubiquitin Binding Entities (TUBES) that bind to K48-linked tetraubiquitin chains to see if the mutation of K730, K759, or K992 in hZO-2 changed the degree of K48 polyubiquitination in ZO-2. For this purpose, MDCK cells were co-transfected with HA-Ubi and Flag_3_-hZO-2-WT or the ubiquitin mutants FLAG_3_-hZO-2-K730R, -K759R, or -K992R. After 24 h, the cells were lysed and treated with a His-tagged anti-K48 TUBE. K-48 poly-ubiquitinated proteins were purified with a His-tag affinity resin, run in an SDS-PAGE, and detected on Western blots with an anti-FLAG antibody. Figure 3 shows that compared to FLAG_3_-hZO-2-WT, K48-polyubiquitination is augmented in FLAG_3_-hZO-2K730R SUMOylation mutant and diminishes in the mutants FLAG_3_-hZO-2-K759R and -K992R, thus indicating that residues K759 and K992 of hZO-2 are indeed targets of K48 poly-ubiquitination. In contrast, the elimination of the K730 SUMOylation site in hZO-2 promotes the poly-ubiquitination of the protein with K48-chains.

### 3.3. The Turnover Rate of hZO-2 Is Regulated by the K48 Polyubiquitination Sites K759 and K992 and by the SUMOylation Site K730

Next, we analyzed if the turnover of hZO-2 was modulated by residues K730, K759, and K992. For this purpose, we transfected MDCK monolayers with Flag_3_-hZO-2-WT or the mutants FLAG_3_-hZO-2-K730R, -K759R, or -K992R, and 6 h after transfection, protein synthesis was blocked with 30 μM cycloheximide. Then, we analyzed the decay of the transfected proteins at different time points. Western blot analyses revealed that the decay of ZO-2 is slowed down in the FLAG_3_-hZO-2 ubiquitination mutants K759R or K992R compared to Flag_3_-hZO-2 WT (Figure 4A,B,D). Furthermore, the calculus of the half-lives for hZO-2 WT was 19.9 h, whereas that of hZO-2 ubiquitination-site mutants K759R or K992R increased to 37.3 h and 23.3 h, respectively (Figure 4E). These results hence indicate that residues K759 and K992, which are K48 poly-ubiquitination target sites, reduce the stability of hZO-2. 

SUMOylation is proposed to exert the opposite effect of ubiquitination regarding protein stability [33]. Accordingly, we observed that the lack of SUMOylation target residue K730 accelerates the decay of hZO-2 (Figure 4C,D) and leads to a reduced half-life of 6.7 h (Figure 4E).

Our results indicate that K48-poly-ubiquitination of residues K759 and K992 and the SUMOylation of K730 regulate hZO-2 stability.

### 3.4. TER Development Diminishes and Paracellular Permeability Augments in ZO-2 Mutants Lacking Critical Ubiquitination or SUMOylation Target Residues

Inhibiting the degradation of proteins in the proteasome augments the cellular content of defective proteins in tissues [34,35]. Therefore, we wondered if the presence of ZO-2 lacking critical residues for K48 poly-ubiquitination could have a deleterious effect on the sealing of TJs. For this purpose, we plated MDCK cells at confluency on Transwell inserts and 24 h later transfected the monolayers with Flag_3_-hZO-2-WT or the ubiquitination mutants Flag_3_-hZO-2 —K759R or —K992R in DMEM without calcium and serum. After 6 h, the monolayers were transferred to DMEM with calcium and serum, and the reestablishment of transepithelial electrical resistance (TER) was measured continuously. Figure 5A shows that monolayers transfected with mutated hZO-2-K759R and -K992R reached a lower peak of TER in comparison to monolayers transfected with WT hZO-2. This result suggests that establishing a TJ with a ZO-2 protein that cannot be adequately ubiquitinated is deleterious due to the presumptive maintenance of defective ZO-2 molecules in the junction. In addition, we also observed that monolayers of MDCK cells transfected with FLAG_3_-hZO-2-K730R that lacks a SUMOylation target residue exhibit a lower value of peak TER. This observation suggests that reducing the half-life of the hZO-2-K730R mutant is deleterious for TJ sealing, probably due to diminished protein stability. 

To further confirm that the lack in hZO-2 of these critical residues for ubiquitination and SUMOylation alters TJ barrier function, we next measured the paracellular permeability of the monolayers with Flag3-hZO-2-WT, the ubiquitination mutants Flag3-hZO-2-K759R or -K992R, and the SUMOylation mutant FLAG3-hZO-2-K730R. Figure 5B shows that in comparison to Flag3-hZO-2-WT, both ubiquitination mutants Flag3-hZO-2-K759R and -K992R, as well as the SUMOylation mutant Flag3-hZO-2-K730R displayed an increase in the paracellular flux of FITC-Dextran of 10 kDa or 70 kDa. These results hence show that these ubiquitination and SUMOylation target sites are important to regulate the sealing of TJs. 

### 3.5. Mutations in ZO-2 of Critical Ubiquitination and SUMOylation Sites Perturb the Interaction with ZO-1 and Occludin

The decrease in TER and the increase in the paracellular flux observed in monolayers transfected with ubiquitination and SUMOylation mutants could be due to an altered interaction of ZO-2 with TJ proteins. Therefore, we transfected MDCK monolayers with Flag_3_-hZO-2-WT or the mutants FLAG_3_-hZO-2-K759R, -K992R, and -K730R. One day later, transfected hZO-2 was immunoprecipitated with antibodies against the FLAG-tag, and the interacting ZO-1 and occludin proteins were detected on Western blots with specific antibodies against these proteins. Figure 6 reveals that the lack of K759 and K992 enhances ZO-2 interaction with occludin and diminishes the association with ZO-1. In contrast, the hZO-2-K730R mutation revealed an increased interaction of hZO-2 with ZO-1 and a decreased association with occludin. 

## 4. Discussion

ZO-2 is a scaffold protein that moves between the cytoplasm, the nucleus, and the TJs, participating in various functions at the cell border and the nucleus. Here, we have analyzed the impact of two putative ubiquitination sites and one previously confirmed SUMOylation site on ZO-2 stability. First, we analyzed K759 and K992 residues of hZO-2 as these lysines are conserved in ZO-2 proteins derived from different animal species and constitute putative targets for both ubiquitination and SUMOylation according to an in silico analysis. The sites modified by both ubiquitin and SUMO, known as Sites of Alternative Modification (SAMs), have been uncovered in thousands of proteins, and, in fact, a recent report indicates that out of 36,000 sites known to be modified by SUMO, 51.8% also are modified by ubiquitin [36]. However, when we previously analyzed the SUMOylation in hZO-2 of site K730 and SAM residues K759 and K992, we found that while the former is strongly SUMOylated, residues K759 and K992 are not [11]. Therefore, we have now explored their role in ZO-2 ubiquitination. 

We observed that hZO-2 K759R and K992R mutants are significantly less ubiquitinated than wild-type ZO-2, whereas the SUMOylation mutant K730R exhibits a higher ubiquitination level. These observations indicate that lysines 759 and 992 of ZO-2 are critical ubiquitination sites in hZO-2. In the future, it will be important to study if both K759 and K992 are simultaneously ubiquitinated in vivo and their relationship with the ubiquitination of several other putative ubiquitination sites present in hZO-2. We also found that the inhibition of a SUMOylation target residue in ZO-2 favors the ubiquitination of the protein. Likewise, in a previous work [11], we observed a minor but not significant increase in SUMOylation of ZO-2-K759R and -K992R mutants. 

We also observed that the half-life of hZO-2 mutant K730R diminished to 6.7 h compared to the 19.9 h detected in wild-type hZO-2. These results agree with the observation that SUMOylation regulates protein stability by antagonizing ubiquitination either in the same lysine as can be the case for SAM residues or by other mechanisms that might include steric hindrance. Enhanced protein stability by SUMOylation has been documented in several proteins, including PI3K p110 catalytic subunit [37], tau [33], phosducin [38], and the transcription factor Nrf2 [39]. However, in the case of the TJ protein claudin-2, its expression level and membrane localization diminished when SUMO-1 was stably expressed [40]. 

Likewise, we demonstrated that ubiquitination target mutants K759R and K992R have a diminished formation of K48-linked ubiquitin chains, the canonical signal that targets proteins to the proteasome for degradation. Instead, in the hZO-2 SUMOylation mutant K730R, K48-linked ubiquitin chains are more abundant. Furthermore, we observed that the K759R mutant increased the half-life of ZO-2 from the 19.9 h observed in the wild type to 37.3 h. A less intense effect was found with the K992R, where the half-life was 23.3 h. Increments in the half-life of these mutants agree with their respective reduced ubiquitination. 

Previously, it was demonstrated that in sparse cultures of MDCK where cells are in proliferation and TJs are only present in cells located in the center of the islets, the half-life of ZO-2 is 8.7 h in comparison to the 19.1 h detected in confluent monolayers [41]. Likewise, we have shown that when TJs are not formed due to the culture of confluent monolayers in low calcium (1–5 μM) condition, ZO-2 displays a diffuse distribution in the cytoplasm and a half-life of 7 h in comparison to the 19.7 h observed in cells cultured in normal calcium (1.8 mM) where ZO-2 concentrates at the TJ [42]. In the low calcium condition, treatment with MG132, a peptide that inhibits the proteasome, augments ZO-2 content, thus indicating that a considerable portion of ZO-2 not associated with the TJ is degraded in the proteasome. However, we also previously showed that a small portion of ZO-2 in the low calcium condition is protected from proteasomal degradation due to its association with 14-3-3 proteins [42]. When TJ assembly is triggered by transferring the epithelial monolayers from a low calcium media to normal calcium media in a protocol known as Ca^2+^-switch, ZO-2 dissociates from 14-3-3 and incorporates into the TJ at the cell borders. With the Ca^2+^-switch, the amount of ZO-2 diminishes compared to the low calcium condition due to lysosomal degradation in a process mediated by endocytosis [42]. These observations suggest that when TJs are de novo assembled, the number of ZO-2 binding sites at the cell borders is limited, inducing the endocytosis and posterior degradation at the lysosomes of excessive ZO-2 inserted at the apical junctional complex. Besides ZO-2, other cytoplasmic junctional proteins like δ-catenin are degraded by both the ubiquitin–proteosome pathway and the lysosome degradation route [43].

In confluent MDCK cells cultured in normal calcium media, ZO-2 is 22-fold more concentrated in the TJ than in the cytoplasm. Junctional ZO-2 is rapidly exchanged with the cytoplasmic pool since the recovery of ZO-2 fluorescence after photobleaching shows a t_½_ of 126 s, even faster than the 161 s found for ZO-1 [44]. ZO-2 ubiquitination mutants K759R and K992R show a significant decrease in the formation of K48-linked ubiquitin chains which prompt proteasomal degradation. Therefore, the half-life increase of ZO-2 ubiquitination mutants K759R and K992R might be partly due to a low turnover rate of the cytoplasmic pool from which ZO-2 is constantly inserted into the junction. In the future, however, it will be important to explore if K63-linked ubiquitin chains that trigger endocytosis and lysosomal degradation [45] are formed in K759 and K992 residues of ZO-2. Especially when a Ca^2+^-switch de novo assembles TJs, and the excess of ZO-2 at the cell borders is endocytosed and degraded in lysosomes [42].

Since ubiquitination is a post-translational modification that regulates endogenous protein stability, it has become important to study its impact on cell–cell junctions. Here, we observed that the hZO-2 mutant K759R, which displays the most substantial decrease in ubiquitination and the highest increase in the half-life of all the mutants tested, does not display the peak of TER that is observed in newly plated cells. It remains unclear why MDCK cells exhibit this initial peak of resistance, but a lack of it has also been observed in ZO-2 KD cells [46,47]. This observation suggests that blocking hZO-2 ubiquitination in K759 allows the insertion of aged/faulty molecules of ZO-2 from the cytoplasm into the TJ. It is also pertinent to mention that mouse ZO-2 K764, the corresponding lysine of K759 in hZO-2, is a target residue for a covalent modification by the organophosphate pesticide methamidophos in mice testis [48]. This pesticide exerts a deleterious effect on blood–testis barrier (BTB) sealing, probably due to a lower turnover rate of defective ZO-2 molecules. 

Here, we also showed that both hZO-2 ubiquitination mutants K759R and K992R, as well as the SUMOylation mutant -K730R increased the paracellular flux of 10 kDa or 70 kDa dextrans. These results further highlight the importance of these ubiquitination and SUMOylation target sites for the sealing of TJs. In addition, it is noteworthy that the hZO-2 ubiquitination and SUMOylation mutants also displayed an altered interaction with the TJ proteins occludin and ZO-1. This observation is also in line with a previous report showing that the ubiquitination of E-cadherin, an adherens junction protein, alters the formation of the adhesion complexes by reducing E-cadherin-binding to p120 [49]. 

Regarding the ubiquitination of other TJ proteins, occludin [50] and tricellulin [51] have been found to bind to the E3-Ub-ligase Itch which promotes their ubiquitination and degradation. In retinal endothelial cells, VEGF induces occludin phosphorylation at S490 by PKCβ, leading to occludin ubiquitination, endocytosis, and blood–retinal barrier (BRB) disruption [52,53]. Ischemia also triggers the ubiquitination of occludin in the BRB [54] and in the blood–brain barrier, where the participation of Itch is involved [55]. Itch also regulates occludin ubiquitination and TJ dynamics in the BTB [56]. Other E3-Ub-ligases involved in occludin ubiquitination are Nedd4-2 in the kidney collecting duct [57] and MARCH3 in endothelia [58], as well as the E2-Ub-conjugating enzyme Ube2j1 in Sertoli cells [59]. 

Claudin stability is also regulated by ubiquitination. Thus, claudin-5 polyubiquitination on K199 triggers its proteasome-dependent degradation [60], and AMPK activation promotes the phosphorylation of claudin-1 in Thr191, blocking the mono-ubiquitination of the protein in residue K189, leading to the enlargement of TJ areas and an increased TER [61]. The E3-Ub-ligase LNX1p80 promotes the ubiquitination and posterior endocytosis and lysosomal degradation of claudin-1, the endocytosis and degradation of claudin-2, and the degradation of claudin-4 [62]. In the distal tubules of the kidney, the E3-Ub-ligase KLHL3 ubiquitinates and degrades claudin-8. The interaction of the latter with claudin-4 is crucial crucial to form a paracellular chloride channel that, when affected, leads to electrolyte and blood pressure imbalances like pseudohypoaldosteronism type II [63]. In the thick ascending limb of Henle, the E3-Ub-ligase PDZRN3 mono-ubiquitinates and triggers the endocytosis of dephosphorylated claudin-16 that forms a paracellular channel for Mg^2+^ reabsorption [64]. In the case of ZO-1, Japanese encephalitis viral infected astrocytes induce the expression of VEGF, IL-6, and MMP2/MMP-9 that upregulate the expression of the E3-Ub-ligase component n-recognin-1 that triggers ZO-1 ubiquitination and degradation [65]. No Ub-conjugating enzymes have yet been reported to interact with ZO-2 and ZO-3.

In summary, we have identified residues K759 and K992 in hZO-2 as acceptors of ubiquitination and targets of K48 poly-ubiquitination. We found that compared to wild-type hZO-2, the half-life of hZO-2 ubiquitination mutants K759R and K992R augmented, while that of the SUMOylation mutant K730R diminished. Mutagenesis of these lysines in hZO-2 affected the interaction of ZO-2 with other TJ proteins, the development of TER and the paracellular flux, highlighting the importance of ubiquitination and SUMOylation for maintaining a healthy pool of ZO-2 proteins in epithelial cells.

## Figures and Tables

**Figure 1 cells-11-03296-f001:**
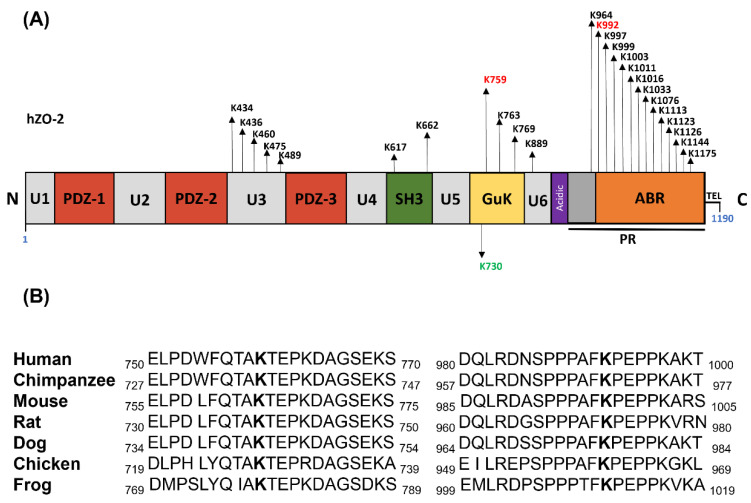
Residues K759 and K992 in hZO-2 are potential targets of ubiquitination. (**A**) Schematic representation of putative ubiquitination sites present in hZO-2. Putative ubiquitination sites are indicated above the scheme of hZO-2, and residues K759 and K992 are shown in red, whereas the SUMOylation site K730 is shown below in green. The scheme of hZO-2 illustrates the distribution of PDZ, SH3, Guk (guanylate kinase), and acidic domains; the unique (U) regions 1 to 6; the proline-rich (PR) region; the actin-binding region (ABR); and the PDZ-binding motif, TEL. (**B**) Residues K759 and K992 are conserved in different species. K759 and K992 residues are marked in bold in the alignment of ZO-2 amino acid sequences from different species.

**Figure 2 cells-11-03296-f002:**
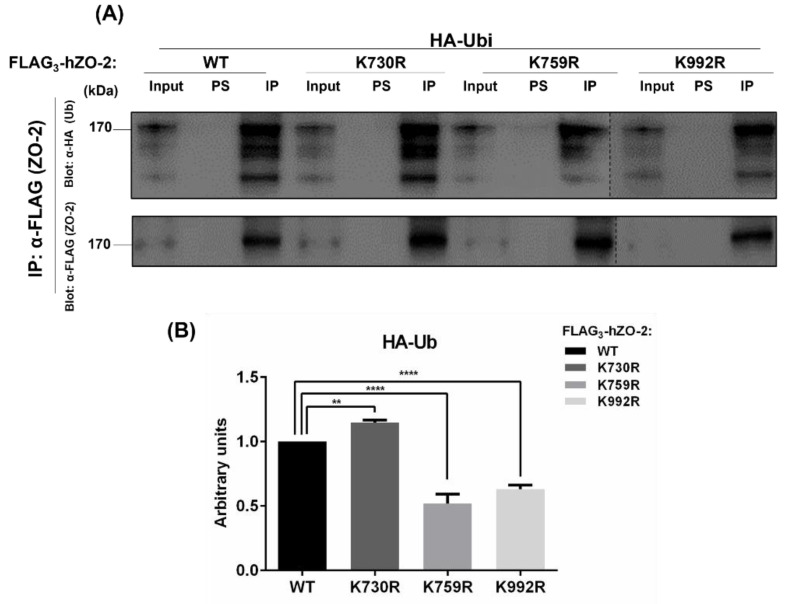
Residues K759 and K992 of hZO-2 are ubiquitination targets. MDCK cells were co-transfected with HA-ubiquitin (HA-Ubi) and Flag_3_-hZO-2-WT or the mutants FLAG_3_-hZO-2-K759R, -K992R, or -K730R. Cell lysates were immunoprecipitated (IP) with anti-FLAG antibody or pre-immune serum (PS), and the precipitates were analyzed by Western blotting with anti-HA (Ubi) and anti-Flag (ZO-2) antibodies. Upper panel (**A**), representative Western blot; lower panel (**B**), quantitative analysis of three independent experiments. Statistical analysis was done with One-way ANOVA followed by Bonferroni’s multiple comparison test; ** *p* < 0.01, **** *p* < 0.0001.

**Figure 3 cells-11-03296-f003:**
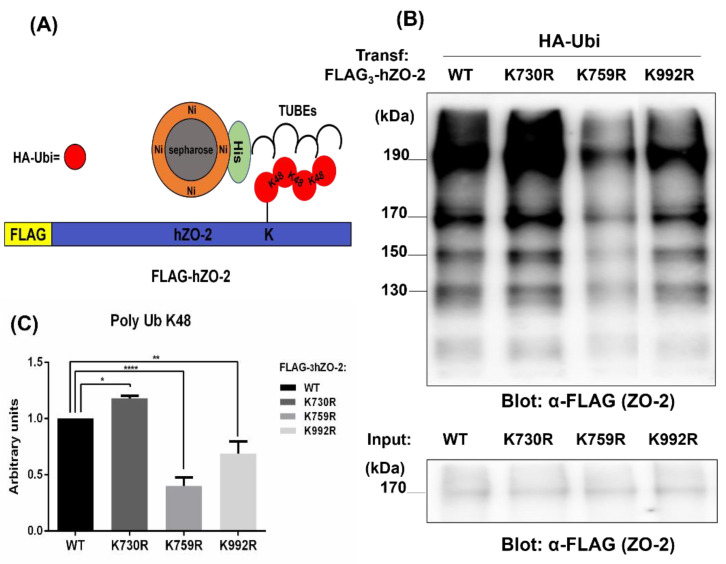
K48-poly-ubiquitination is increased in SUMOylation-defective hZO-2 mutant K730R and diminishes in hZO-2-K759R and K992R mutants. MDCK cells were co-transfected with HA-Ubi and Flag_3_-hZO-2 WT, and the ubiquitination mutants FLAG_3_-hZO-2-K759R or -K992R or the SUMOylation mutant FLAG_3_-hZO-2-K730R. After 24 h, the cells were lysed and treated with a His-tagged anti-K48 TUBE. K-48-poly-ubiquitinated proteins were purified with a His-tag affinity resin, run in an SDS-PAGE, and blotted against FLAG. (**A**) Schematic representation of Flag_3_-hZO-2 where K48-poly-ubiquitination associates to TUBEs that bind to a Ni^2+^-chelate matrix through a His-tag. (**B**) Representative Western blot of cell lysates. (**C**) Quantitative analysis of the area in the blot containing the bands of 190, 170, 150, and 130 kDa. Data from three independent experiments. Statistical analysis was done with One-way ANOVA followed by Bonferroni’s multiple comparison test; ns, non-significant, * *p* < 0.05, ** *p* < 0.01, **** *p* < 0.0001.

**Figure 4 cells-11-03296-f004:**
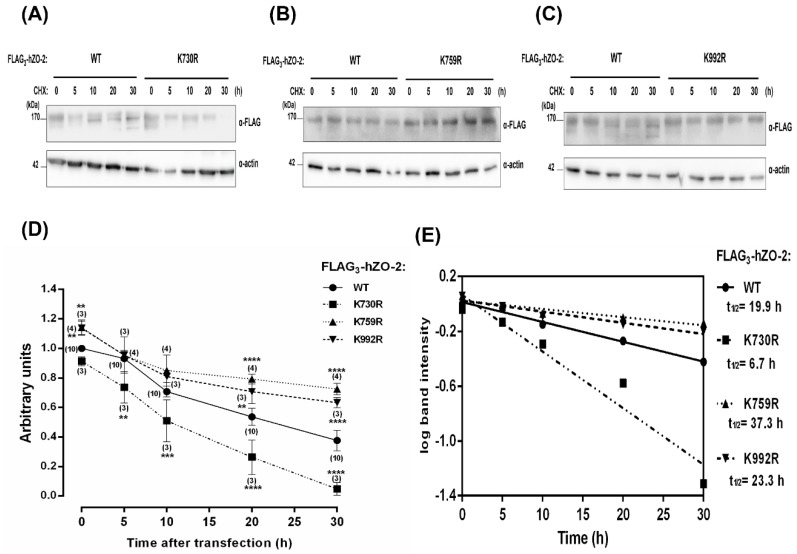
The stability of hZO-2 is regulated by K48 poly-ubiquitination sites K759 and K992 and by the SUMOylation site K730. MDCK cells were transfected with Flag_3_-hZO-2 WT, the ubiquitination mutants FLAG_3_-hZO-2 K759R and K992R, or the SUMOylation mutant FLAG_3_-hZO-2 K730R. After 6 h, protein synthesis was blocked with 30 μM cycloheximide for the indicated times (h). (**A**–**C**) Representative images of cell lysates analyzed on Western blots with antibodies against FLAG or actin, latter used as a loading control. (**D**) Quantitative analysis of Western blots derived from three independent experiments. Statistical analysis was done with a Two-way ANOVA followed by Bonferroni’s test. ** *p* < 0.01, *** *p* < 0.001, **** *p* < 0.0001. (**E**) Half-life determination. Measured half-lives are 19.9 h for hZO-2 WT, 6.7 h for hZO-2 K730R, 37.3 h for hZO-2 K759R, and 23.3 h for hZO-2 K992R.

**Figure 5 cells-11-03296-f005:**
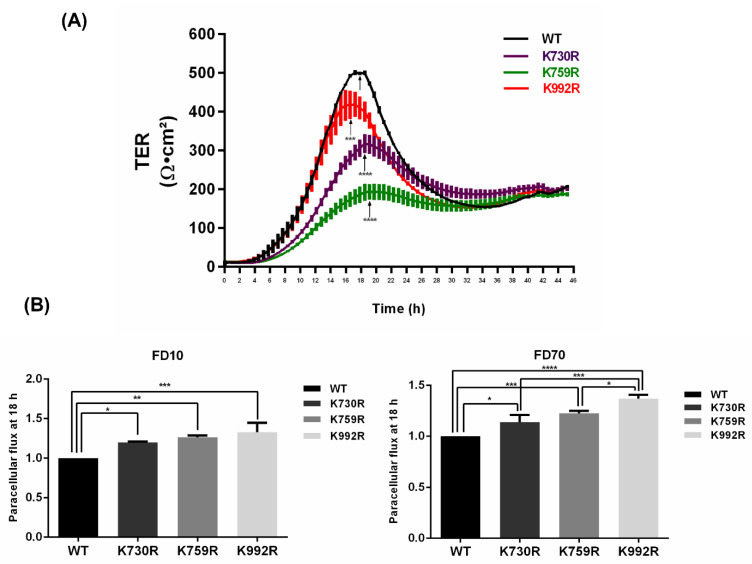
The expression of ZO-2 mutated in critical ubiquitination or SUMOylation target residues impairs TJ barrier function. MDCK monolayers were plated on Transwell inserts and one day later were transfected with Flag_3_-hZO-2 WT, the ubiquitination mutants FLAG_3_-hZO-2-K759R and -K992R, or the SUMOylation mutant FLAG_3_-hZO-2-K730R in DMEM without calcium and serum. After 6 h, monolayers were switched to DMEM with calcium and serum, and the TER values were continuously recorded using the CellZscope system after that. (**A**) Monolayers transfected with hZO-2 ubiquitination and SUMOylation mutants exhibit a lower peak of TER than those transfected with Flag_3_-hZO-2 WT. The graph presents the mean TER ± SD from 3 monolayers plated on Transwell inserts. The graph is representative of three independent experiments. Statistical analysis was done with One-way ANOVA followed by Bonferroni’s multiple comparison test, comparing the peak values of TER (arrows) of each tested mutant with that of Flag_3_-hZO-2 WT. *** *p* < 0.001, **** *p* < 0.0001. (**B**) Monolayers transfected with hZO-2 ubiquitination and SUMOylation mutants display at 18 h after the Ca-switch a higher paracellular passage of 10 kDa FITC-Dextran (FD10) and 70 kDa FITC-Dextran (FD70) measured in the apical to basolateral direction than those transfected with Flag_3_-hZO-2 WT. The graphs present the mean ± SD of three independent experiments. Statistical analysis was done with One-way ANOVA followed by Bonferroni’s multiple comparison test. Values shown are normalized to control (Flag_3_-hZO-2 WT). * *p* < 0.05, ** *p* < 0.01, *** *p* < 0.001, **** *p* < 0.0001.

**Figure 6 cells-11-03296-f006:**
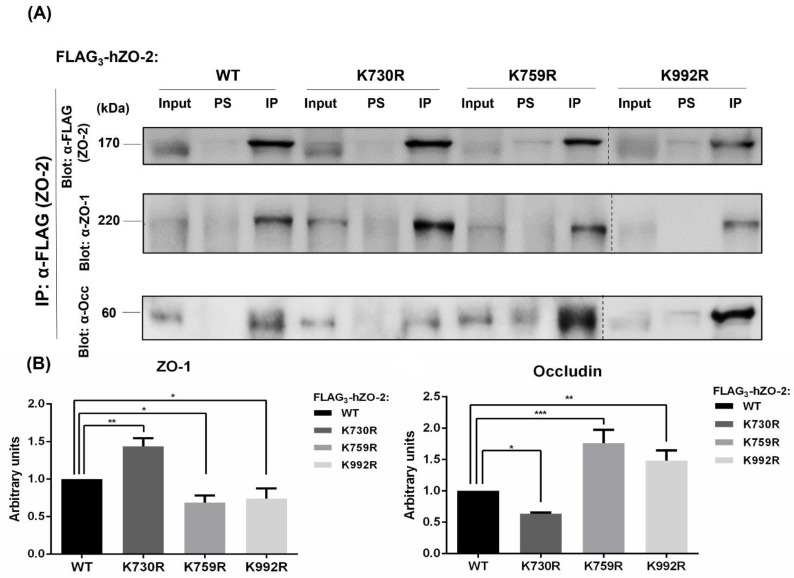
The mutation of K759 or K992 ubiquitination sites, or K730 SUMOylation site in hZO-2 alters the interaction with occludin and ZO-1. MDCK monolayers were transfected with Flag_3_-hZO-2 WT or the mutants FLAG_3_-hZO-2 -K759R, -K992R, and -K730R. Transfected hZO-2 was immunoprecipitated with antibodies against FLAG, and the interacting ZO-1 and occludin were detected with specific antibodies directed against these proteins. (**A**) Representative images of Western blots. (**B**) Quantitative analysis of Western blots derived from three independent experiments. Statistical analysis was done with a One-way ANOVA followed by Bonferroni’s test. * *p* < 0.05, ** *p* < 0.01, *** *p* < 0.001.

## Data Availability

Not applicable.

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
