# Peer review of "Polyubiquitination and SUMOylation Sites Regulate the Stability of ZO-2 Protein and the Sealing of Tight Junctions"

_cells, 2022, doi:10.3390/cells11203296_

Round 1
Reviewer 1 Report
This manuscript investigates the effect of ZO-2 ubiquitination and SUMOylation on ZO-2 protein stability and the tight junction barrier function by using ubiquitination-site mutants hZO-2 -K759R, -K992R or -K730R. The experiments were all well-designed and well-executed. The manuscript is clearly presented and easy to follow. One comment is about figure 4 (see below). Other than that, this study is significant and provides very important information about ZO-2 regulation and stability in the cells.
Specific comment:
Figure 4: The authors indicate that K730R did not affect ZO-2 binding with occludin in Figure 4B (occludin). However, from the Western blot result shown in figure 4A, it looks like the occludin signal is reduced in K730R. The authors should provide some explanation about this.
Reviewer 2 Report
The authors identified that residues K759 and K992 in hZO-2 are acceptors for ubiquitination. Moreover, residues K759 and K992 of hZO-2 are targets of K48 polyubiquitination. Accordingly, the half-life of hZO-mutants -K759R and -K992R augmented from 19.9 to 37.3 and 23.3 h, respectively. Instead, the ubiquitination of hZO-2 mutant K730R increased, and its half-life diminished to 6.7 h. The lack of these lysine residues in hZO-2 affects TJ sealing as the peak of TER decreased in monolayers of MDCK cells transfected with any of these mutants. They concluded that ZO-2 ubiquitination and SUMOylation is important to maintain a healthy and stable pool of ZO-2 molecules at the TJ. Overall, the current result is not solid enough to support the conclusion; the molecular mechanism was not well studied.
I have the following concerns over the paper, which requires a major revision.
1. K730 is not shown in figure 1A.
2. There is no or too weak band in Input group in figure 2A and 6A.
3. The authors need to construct K759 and K992 double mutant for confirmation.
4. The actin band is not equal among various groups in figure 4A-C.
5. The authors identified the role of ZO-2 ubiquitination in TJ sealing via measurement of TER, is there any other method to confirm the phenomenon? As both ubiquitination sites, why K992R showed much higher TER than K759R? How these sites regulate the sealing of tight junction, what is the molecular mechanism? As important content of the paper, more work needs to be done to achieve the conclusion.
6. Why there are many bands of ZO-1 and ZO-2 in figure 2A and 6A, but one band in figure 4A?
7. Is there any effect of ubiquitination on its SUMOylation?
Round 2
Reviewer 2 Report
The authors have answered most questions and comments satisfactorily. However, K759 and K992 double mutant still looks essential for confirmation.
.
